# Lipid Oxidation of Stored Brown Rice Changes Ileum Digestive and Metabolic Characteristics of Broiler Chickens [note 1]

**DOI:** 10.3390/ijms26147025

**Published:** 2025-07-21

**Authors:** Beibei He, Xueyi Zhang, Weiwei Wang, Li Wang, Jingjing Shi, Kuanbo Liu, Junlin Cheng, Yongwei Wang, Aike Li

**Affiliations:** 1Academy of National Food and Strategic Reserves Administration, Beijing 100037, China; hbb@ags.ac.cn (B.H.); nancheng0710@163.com (X.Z.); www@ags.ac.cn (W.W.); wl@ags.ac.cn (L.W.); sjj@ags.ac.cn (J.S.); lkb@ags.ac.cn (K.L.); cjl@ags.ac.cn (J.C.); 2School of Food Science and Engineering, Wuhan Polytechnic University, Wuhan 430048, China

**Keywords:** stored brown rice, lipid oxidation, meat quality, antioxidant and digestible enzyme, metabolic characteristics, broiler chickens

## Abstract

Long-term storage may induce lipid oxidation in brown rice and impact its utilization in animal diets. One-day-old male Ross 308 broiler chickens (with an initial body weight of 20 g) were randomly divided into three groups: corn-based diet (Corn), fresh brown rice-based diet (BR1) and stored brown rice-based diet (BR6), with 8 replicates of 10 birds per pen, in a 42-day feeding trial. The results showed that lipid oxidation indexes increased and fatty acid composition changed significantly in BR6 (*p* < 0.05). The dietary replacement of corn with brown rice showed no effects on growth performance of broilers (*p* > 0.05). However, palmitic acid and oleic acid increased, and stearic acid, linoleic acid and docosadienoic acid decreased in the broiler breast muscle of the BR1 and BR6 groups (*p* < 0.05). Ileum antioxidant enzyme activities increased in the BR1 and BR6 groups compared to the Corn group (*p* < 0.05), and the activities of α-amylase, trypsin, chymotrypsin and lipase decreased in the BR6 group compared to the BR1 and Corn groups (*p* < 0.05). Also, compared to the BR1 group, the overall expression of metabolites involved in drug metabolism—cytochrome P450, GnRH secretion and the estrogen signaling pathway in broiler ileum were down-regulated in the BR6 group (*p* < 0.05). In conclusion, the lipid oxidation of stored brown rice decreased digestive enzyme activities and changed metabolic characteristics in the ileum of broilers. While replacing corn with brown rice did not affect broiler growth performance, it reduced the contents of unsaturated and essential fatty acids in breast muscle and enhanced the ileal antioxidant functions of broilers.

## 1. Introduction

According to the latest forecasts from the FAO, global rice stocks are expected to reach a historical high of 204 million tons by the end of the 2024/25 marketing year. As the largest holder of rice reserves globally, the Chinese government has been conducting annual auctions of old paddy rice for feed use in recent years, which offers alternative grain feedstuff and mitigates the upward pressure on feed prices. Brown rice is a product of husked paddy rice which retains its bran and germ, and is high in dietary fiber, vitamins and minerals [1]. The nutrient value of brown rice is comparable to or higher than that of corn, and could achieve a similar feeding effect when used as an alternative ingredient in animal feed [2]. However, the inherent respiration, oxidation and microbial activity during storage could alter the composition and structure of its key nutrients, thereby degrading the quality of brown rice [3].

Although the lipid content in brown rice is only 1–3%, it undergoes significant changes during the longtime storage, primarily through hydrolysis and oxidation pathways. Lipids are firstly hydrolyzed by lipases and phospholipases, generating free fatty acids and glycerol [4], mainly including linoleic acid, palmitic acid, linolenic acid, stearic acid, oleic acid and tetradecanoic acid [5]. Meanwhile, unsaturated fatty acids are further oxidized to form carbonyl compounds, thereby causing a decline in the nutritional value of rice [6]. The lipid oxidation of rice is regulated by both lipid oxidase and antioxidant oxidase systems. Lipid oxidase in rice can catalyze unsaturated fatty acids, such as linolenic and linoleic acid, to produce lipid hydroperoxide, which is then automatically oxidized or degraded by lipid hydroperoxide lyase and lipid hydroperoxide isomerase to produce aldehydes, hydrocarbons, alcohols and other volatile substances [7]. Antioxidant enzymes, including catalase, peroxidase, superoxide dismutase, ascorbate peroxidase and glutathione peroxidase, play an important role in eliminating reactive oxygen species (ROS) in plant cells [8].

Feed oxidation has been identified as an important factor affecting the growth performance of animals, which may reduce feed palatability and feed intake [9]. Moreover, it has been demonstrated that oxidation can diminish feed nutritional quality, as it triggers a reaction with proteins, lipids and fat-soluble vitamins present in the diet [10]. Certain oxidation byproducts, like unsaturated aldehydes, can cause damage to the liver and intestines of animals, even at low concentrations [11]. Tan et al. supplemented 4% oxidized fish oil in the diet of female broilers for 21 days, and found that the oxidized fish oil group showed a higher hepatic malondialdehyde (MDA) concentration and the down-regulation of tight junction proteins (claudin-1 and occludin), indicating enhanced lipid per-oxidation in the liver and impaired intestinal barrier function [12]. Lipid oxidation may also harm animal health through a variety of other pathways, including inhibiting hepatic P450 enzyme activity and disrupting mitochondrial β-oxidation [13,14]. Previous studies have also shown that lipid peroxides can induce oxidative stress, leading to damage to membrane integrity and protein cross-linking/denaturation, which induce the decreased water-holding capacity and pH of the muscle [15]. As metabolomics can reveal how lipid oxidation products interact with metabolic pathways in animals and how these changes affect the physiological state of animals, it can provide important clues for understanding the potential impact of lipid oxidation on animal health [16].

The existing research on the impact of feed oxidation on animals mainly focuses on oil, protein feed ingredients, vitamins and minerals, as well as feed additives that are prone to oxidation [17,18]. However, as grains are the component that is added to feed in the highest amount, most studies only focus on changes in energy value, nutrient digestibility and the effects of long-term stored grains on animal growth performance, while few pay attention to the changes in animal physiological metabolism caused by grain lipid oxidation; therefore, in-depth research is lacking at the metabolic pathway level. Therefore, the present study aimed to systematically explore the comprehensive impact of lipid oxidation in brown rice on the growth performance and physiological metabolism of animals, particularly by revealing potential changes in metabolic pathways through metabolomics methods, which may provide a theoretical basis for the rational application of stored brown rice in broiler chickens.

## 2. Results

### 2.1. Lipid Metabolism Indexes and Fatty Acid Composition Changes of Stored Brown Rice

The comparison of various lipid metabolism indexes between fresh brown rice (BR1) and stored brown rice (BR6) are presented in Table 1. The BR6 generally showed higher levels of lipid peroxidation markers, which include fatty acid value, malonaldehyde (MDA) and carbonylated protein, compared to that of BR1 (*p* < 0.05). In the meantime, the BR6 exhibited lower levels of glutathione peroxidase (GSH-PX) and glutathione (GSH) compared to the BR1 (*p* < 0.05). There were no significant differences in superoxide dismutase (SOD), catalase (CAT), lipase (LPS) and ascorbate peroxidase (APX) between the BR1 and BR6 (*p* > 0.05). These results indicate that the BR1 possesses greater oxidative stability and stronger antioxidant capacity than the BR6, which are essential for maintaining both the quality and the nutritional value of brown rice.

Fatty acid compositions of fresh and stored brown rice are shown in Table 2. Compared with the BR1, BR6 exhibited lower contents of linoleic acid (C18: 2n-6c), alpha-linolenic acid (C18: 3n-3) and tricosanoic acid (C23: 0), but higher levels of stearic acid (C18: 0), oleic acid (C18:1n-9c), icosanoic acid (C20: 0), eicosenoic acid (C20: 1), docosanoic acid (C22: 0) and tetracosanoic acid (C24: 0) (*p* < 0.05). The content of total fatty acid and the sum of saturated fatty acids (SFAs) showed no differences between the BR1 and BR6 (*p* > 0.05). The content of monounsaturated fatty acids (MUFAs) was higher, and that of polyunsaturated fatty acids (PUFAs) was lower in the BR6 compared to BR1 (*p* < 0.01). The PUFA/SFA ratio was higher in the BR1 than in the BR6 (*p* < 0.01), suggesting that the BR1 possesses a more favorable fatty acid profile that may confer greater health benefits.

### 2.2. Effects of Stored Brown Rice on Growth Performance and Meat Quality of Broilers

In Table 3, no significant differences were determined in average daily feed intake (ADFI) and average daily gain (ADG) from wk 0 to 3, wk 4 to 6, or wk 0 to 6 between broilers fed corn- or brown rice-based diet (*p* > 0.05). Similarly, the feed conversion ratio (FCR) did not differ among dietary treatments at any time point (*p* > 0.05), except for the first 3 wks, when the BR6 group exhibited a higher FCR than that of the Corn and BR1 groups (*p* < 0.05). On the part of meat quality, the pH values were not significantly different among the dietary treatments (*p* > 0.05). However, a significant difference in drop loss was detected, as the BR6 group showed a higher percentage of drop loss compared to the Corn and BR1 groups (*p* < 0.01). The cooking percentage also tended to differ, with the Corn group achieving a higher percentage than the BR1 and BR6 groups (*p* = 0.06). Overall, despite a few trends, most parameters remained statistically similar, indicating that the diet treatments exerted comparable effects on broiler growth performance and meat quality.

The fatty acid compositions of broiler breast muscle were presented in Table 4. Compared with the Corn group, broilers fed two brown rice-based diets showed higher levels of palmitic acid (C16: 0), palmitoleic acid (C16: 1), oleic acid (C18: 1n-9c) and total fatty acid (*p* < 0.05), but lower levels of stearic acid (C18: 0), linoleic acid (C18: 2n-6c), eicosandienoic acid (C20: 2), docosadienoic acid (C20: 4n-6) and docosahexaenoic acid (C22: 6n-3) (*p* < 0.05). Also, the values of PUFAs, PUFA/SFA and EFAs were lower (*p* < 0.05), while the values of MUFAs and MUFA/PUFA were higher (*p* < 0.05) in the two brown rice-based diet feeding groups compared with the Corn group. The levels of myristic acid (C14: 0) and nervonic acid (C24: 1) were significantly higher in the BR6 group compared to the BR1 group (*p* < 0.01).

### 2.3. Effects of Stored Brown Rice on Ileal Antioxidant Properties and Digestive Enzyme Activities of Broilers

Table 5 presents the ileal antioxidant and digestive enzyme indices of broilers fed with corn- or brown rice-based diets. MDA level was highest in the Corn group at 0.94 μmol/g, followed by the BR6 group with 0.62 μmol/g, and was lowest in the BR1 group at 0.51 μmol/g. The decreasing trend in MDA levels from the Corn group to the BR1 group suggests that brown rice may confer protection against intestinal lipid peroxidation in broilers. Conversely, ileal antioxidant enzyme activities including SOD, TAOC, GSH-PX and CAT were higher in the BR1 and BR6 groups than in the Corn group (*p* < 0.01). Digestive enzyme activities including α-AMY, trypsin and chymotrypsin also differed significantly, with the BR6 group exhibiting the lowest value relative to the Corn and BR1 groups. In addition, ileal LPS level was significantly higher in the BR1 group than in the Corn and BR6 groups (*p* < 0.01).

### 2.4. Effects of Stored Brown Rice on Ileal Metabolomic Characteristics of Broilers

Metabolomic biochemical indexes of ileum in broilers fed corn- or brown rice-based diets were tested. Partial least squares discriminant analysis (PLS-DA) analysis of metabolomic data showed significant clustering of metabolites between the Corn, BR1, and BR6 groups (Figure 1A). Affected metabolites were found after filtering by the t-test (*p* < 0.05) and orthogonal least partial squares discriminant analysis (OPLS-DA) model (variable importance in the projection (VIP) > 1.0). The volcano plot showed that there were 91 metabolites decreased (green) and 38 metabolites increased (red) in the BR1 group compared to the Corn group, while there were 73 metabolites decreased and 24 metabolites increased in the BR6 group compared to the BR1 group (Figure 1B). The individual metabolites in each group were analyzed using MetaboAnalyst 4.0 software, and VIP values of the top 20 differential metabolites are shown in Figure 2C,D. Between the Corn and BR1 groups, 1-aminocyclohexanoic acid, D-proline betaine, stachydrine, cis-EODA, 12-hydroxyoctadecanoic acid, Val-Ala, azelaic acid, (S)-leucic acid, sphingosyl-phosphocholine, 5-hydroxyhexanoic acid, 2-hydroxyhexanoic acid, LPC (20:2/0:0), LPC (0:0/20:2), nicotinic acid and guanidineacetic acid were down-regulated, while folic acid, methyl nicotinic acid, 1-methylxanthine, 7-methylxanthine and γ-aminobutyric acid were up-regulated in the BR1 group, compared to the Corn group (Figure 1C). Between the BR1 and BR6 groups, ritalinic acid, γ-aminobutyric acid, N’-formylkynurenine, 7-ketodeoxycholic acid, oxaceprol, 4-hydroxybenzaldehyde, L-dihydroorotic acid, N-acetylcadaverine, PC (8:0/8:0) and thymidine were down-regulated, while FFA (20:1), cis-EODA, 1-methylguanosine, 2′-O-methylguanosine, heparin, 1-methylinosine, quinoline-4-carboxylic acid, 2-methylguanosine, arachidyl glycine and 2-(dimethylamino) guanosine were up-regulated in the BR6 group, compared to the BR1 group (Figure 1D).

The Venn diagram (Figure 2A) shows the number of shared or unique metabolites among the diet treatments. The number of differential metabolites unique to the Corn vs. BR1 groups and the BR1 vs. BR6 groups were 50 and 29, respectively, with 6 differential metabolites common both in the Corn vs. BR1 groups and the BR1 vs. BR6 groups (7-ketodeoxycholic acid, cis-EODA, heparin, γ-aminobutyric acid, 2-methylguanosine, and 1-methylinosine). In Figure 2B, the differential metabolites are divided mainly into four subclasses. In the subclass 1, 13 metabolites had the highest abundance in the BR1 group and the lowest abundance in the Corn group. In the subclass 2, 25 metabolites had the highest abundance in the BR6 group and the lowest abundance in the Corn group. On the contrary, in the subclasses 3 and 4, 31 and 37 metabolites had the highest abundance in the Corn group and the lowest abundance in the BR6 group.

The differential abundance score (DA score) provides a mean to quantify the collective and average alterations of all metabolites within a specific pathway, and is calculated using the following formula: DA score = (number of up-regulated differential metabolites in the pathway − number of down-regulated differential metabolites in the pathway)/total number of metabolites annotated to the pathway. Figure 2C shows the metabolic pathways significantly enriched between the Corn and BR1 groups. The overall expression of metabolites involved in one carbon pool by folate, GnRH secretion and estrogen signaling pathway tended to be up-regulated, while metabolites involved in linoleic acid metabolism and RRAP signaling pathway tended to be down-regulated in the BR1 group compared to the Corn group. Figure 2D shows the metabolic pathways significantly enriched between the BR1 and BR6 groups, as the overall expression of metabolites involved in drug metabolism-cytochrome P450, GnRH secretion and estrogen signaling pathway tended to be down-regulated in the BR6 group. These findings indicate that substantial variations exist in the metabolic pathways among various treatment groups, which could be associated with the physiological condition and metabolic activities of the broilers.

## 3. Discussion

The lipid oxidation is one of the primary factors leading to the aging of the rice. The fatty acid value is an early indicator of lipid oxidation and can reflect the initial stage of this process, which sheds light on the stability and quality of rice stored over extended periods [5]. MDA, a byproduct of lipid peroxidation, serves as a reliable indicator of oxidative stress and is useful for evaluating the degree of lipid damage. Elevated MDA levels suggest increased lipid peroxidation, which can lead to off-flavors and reduced palatability [19]. Carbonylated protein, resulting from the reaction between lipid peroxidation products and proteins, serves as another important biomarker of oxidative stress [20]. Monitoring these indicators is essential for evaluating the oxidative stability of rice and developing strategies to mitigate lipid oxidation during rice storage [21].

The fatty acid composition showed significant differences between the fresh and stored brown rice (*p* < 0.05). Specifically, BR6 showed reduced contents of linoleic acid and α-linolenic acid, and increased content of oleic acid, which may be attributed to lipoxygenase predominantly operates on linoleic acid as its key substrate, and the increased activity of lipoxygenase resulting in the decrease in linoleic acid and increase in the relative content of oleic acid [22,23]. There are nine other trace fatty acids that showed significant changes, and the content of MUFAs was increased and PUFAs was decreased in the BR6, which is consistent with the results reported in other studies [24,25].

As brown rice can be used as an alternative energy feed to corn [2], we replaced 100% of the corn in the diet with fresh and stored brown rice. The results showed that both the BR1 and BR6 diets had no significant effect on growth performance of the broilers, except for an increase in FCR from wk 0 to 3 of the BR6 group (Table 3). Although FCR difference was marginal in this trial, large-scale application could yield substantial cost and feed savings [26]. The freshness of feed has a significant impact on the early growth performance of the broilers [27]. The alterations in composition and structural of major nutrients during longtime storage may lead to a decrement in the nutritional values of grains, particularly in terms of available energy value and amino acid digestibility, especially when subjected to an inadequate storage condition [18]. Nonetheless, several investigative studies conducted on pig and poultry had indicated that the growth performance of animals exhibited a weak correlation with the alterations in nutritional values of grains that have been stored for prolonged durations [28,29]. The value of drip loss is related to the lipid peroxide extent in the muscle [30]. A previous study showed that decreased catalase and peroxidase activities and increased acidity of fatty acid in stored maize may lead to a lower pH and higher drip loss in breast muscle of broilers [31], which is consistent with our results indicating that the increased contents of lipid oxidation in the stored brown rice induced an increase in drop loss of broiler muscles (Table 3). Meat exhibiting elevated drip loss tends to lose moisture during storage and retail display, yielding a dry surface and dull color. The poor water-holding capacity also adversely affects its tenderness and juiciness [32]. Moreover, high-cooking-loss meat displays inferior chewability and excessive firmness, directly undermining eating quality and dampening consumer purchase intent [33].

The fatty acid composition of the diet can affect muscle fatty acid profile of the animals [34]. According to the present study, broilers fed two brown rice-based diets had higher levels of palmitic acid (C16: 0) and oleic acid (C18: 1n-9c), but lower levels of stearic acid (C18: 0), linoleic acid (C18: 2n-6c) and eicosandienoic acid (C20: 2), compared to those of the Corn group. Also, the PUFAs and EFAs were lower, while the MUFAs and MUFA/PUFA were higher in broiler muscle of two brown rice-based diet feeding groups compared with the Corn group (Table 4). Although the fatty acid composition of corn was not determined in this study, and there are no directly relevant studies comparing the fatty acid composition of broiler muscle between corn and brown rice-based diets, it has been demonstrated that the PUFAs in the corn make up for more than 50% of the total fatty acids, with the SFAs accounting for 17% and MUFAs for another 32%, respectively [35]. In contrast, brown rice has a more balanced distribution of the MUFAs and PUFAs, with oleic acid (C18: 1) and linoleic acid (C18: 2) each accounting for 35% to 40% of the total fatty acids, and the SFAs accounting for 32% [36]. One study has showed that as the content of PUFAs in feed increased from 15 to 61 g/kg, the level of PUFAs in chicken meat also significantly increased, while the levels of SFAs and MUFAs decreased accordingly [37]. The increase in the PUFA content in broiler muscle, especially omega-3, may help meet consumers’ demands for healthier food. However, on the other hand, a higher content of PUFAs can also lead to a decrease in the oxidative stability of the meat, which in turn results in a decline in meat flavor and shortened shelf life [38]. A balanced fatty acid composition in feed, especially a higher content of MUFAs, can enhance the oxidative stability of broiler muscle and reduce the impact of oxidation reactions on meat quality [39]. Although a lower concentration of PUFAs may lead to better oxidative stability of the muscle, the deficiency of essential fatty acids may affect the growth performance and immunomodulatory function of broilers. For example, the decreased content of PUFAs and increased content of SFAs in broiler muscle under heat stress conditions can improve the oxidative stability of the muscle, and it will also lead to the decline in the growth performance and immune function of broilers [40]. Due to the lack of data on the fatty acid composition of corn, the extent to which corn drives the observed shifts in broiler-muscle fatty acids remains uncertain. Future studies should therefore characterize the fatty-acid composition of the corn diet to clarify its specific contribution.

Lipid-oxidized feed may alter the fatty acid composition of broiler muscle, and ultimately leads to meat quality deterioration [41]. A study used aging corn in the diet of laying hens and found that the contents of the oleic acid, docosahexaenoic acid and MUFAs were lower, and the contents of stearic acid and SFAs were higher in egg yolks [42]. The inhibition of Δ-9 desaturase activity in ageing grain might be the reason for the reduced conversion of SFAs like C16: 0 and C18: 0 into their monounsaturated counterparts [43]. When broilers were fed with long-term stored rice bran, the content of C14: 0 in the chicken meat was significantly increased, and the content of C18: 1 was significantly decreased compared to that in the fresh rice bran-fed group. Moreover, the meat of broilers fed with long-term stored rice bran showed higher MDA value during the shelf life [44]. In the present study, the contents of linoleic acid (C18: 2n-6c), α-linolenic acid (C18: 3n-3) and PUFAs were reduced, and oleic acid (C18: 1n-9c) and MUFA were increased in brown rice stored for 6 years, compared to fresh brown rice. The result of these changes was that the levels of myristic acid (C14: 0) and nervonic acid (C24: 1) were higher in broilers fed a stored brown rice-based diet compared to those fed a fresh brown rice-based diet. As oxidized feed can promote the synthesis of myristic acid (C14: 0) and inhibit the β -oxidation of nervonic acid (C24: 1) in the liver of broilers by activating the SREBP-1c pathway, it leads to an increase in the contents of these two fatty acids in the muscle [45].

The level of MDA in broiler ileum was significantly higher in the Corn group compared to the two brown rice-based diet feeding groups, and the activities of several key antioxidant enzymes, including SOD, TAOC, GSH-PX and CAT, were significantly lower in the broilers fed a corn diet. Lipid oxidation is a complex process and produces ROS along with other harmful byproducts [46]. As the content of PUFAs in corn is much higher than that in brown rice, it may not only lead to a decrease in the oxidative stability of the muscle, but may also cause oxidative stress in the intestine of broiler chickens [45]. The up-regulation of antioxidant enzymes suggests that the broilers were able to mount a compensatory response to counteract the increased oxidative stress. This adaptive mechanism is indispensable for maintaining cellular homeostasis and minimizing the detrimental effects of oxidative damage on growth and health. However, it is important to note that while the enhanced antioxidant defense mechanisms may help mitigate some of the oxidative stress, prolonged exposure to oxidized feed could still exert negative influences on broiler growth performance and health. The present study provides valuable insights into the effect of lipid oxidation in long-term stored brown rice on antioxidant status in broilers. Specifically, the level of MDA was found to be significantly higher in broilers fed a long-term stored brown rice diet compared to those fed a fresh brown rice diet. Although the activities of antioxidant enzymes were not different between stored and fresh brown rice, the elevated MDA levels in the meat and intestine of broilers fed a long-term stored brown rice diet indicate that the lipid oxidation of prolonged storage brown rice may in turn induce higher oxidative stress in broilers.

The activities of digestive enzymes like α-amylase, trypsin, and chymotrypsin were also affected, as these enzyme activities were lower in the BR6 group compared to the BR1 group (Table 5), indicating that feeding long-term stored brown rice may affect the digestion of starch and protein in broilers, which is consistent with previous research [47]. Although we did not assess the nutrient digestibility of brown rice in broilers, our earlier study in weaned piglets showed that although long-term stored brown rice reduced the activity of lactase in the small intestine of pigs, it had no significant effect on nutrient digestibility and growth performance of piglets [48].

The metabolomic data provided valuable insights into the effects of long-term stored brown rice on the ileum metabolite profiles of broilers. Figure 1C,D present the VIP scores for metabolites that are significantly altered in the Corn group vs. the BR1 group, as well as the BR1 group vs. the BR6 group, respectively. Between the Corn and BR1 groups, folic acid, methyl nicotinic acid, 1-methylxanthine, 7-methylxanthine and γ-aminobutyric acid were up-regulated in the BR1 group compared to the Corn group (Figure 1C). As is known, the brown rice has higher levels of folic acid than the corn, and the increased folic acid in the ileum of brown rice-fed broilers may play crucial roles in various biological processes, including methylation reaction, amino acid metabolism, and DNA synthesis [49]. The brown rice also has a relatively high niacin content compared to the corn, and is mainly in the free form, which is more easily absorbed and utilized by the organism [50]. And the higher content of methyl nicotinic acid in the ileum of brown rice-fed broilers, a derivative of niacin, highlights the potential for enhanced niacin metabolism, which is essential for energy production and cellular respiration, contributing to the maintenance of healthy gastrointestinal function [51]. 1-Methylxanthine and 7-Methylxanthine are derivatives of xanthine, a purine base involved in nucleic acid metabolism. Their up-regulation in the ileum of brown rice-fed broilers indicates a potential increase in purine metabolism, which is essential for DNA and RNA synthesis, and reflects improved cellular function and overall metabolic activity in the gastrointestinal tract [52]. The elevated level of γ-aminobutyric acid (GABA) in the ileum of brown rice-fed broilers is noteworthy. GABA is not only a vital inhibitory neurotransmitter in the central nervous system, but also has significant roles in the gastrointestinal tract, as it can modulate gut motility, reduce inflammation and enhance gut barrier function. The up-regulation of GABA in the ileum suggests that brown rice may exert beneficial effects on gut health by promoting a more favorable gastrointestinal environment [53], which may contribute to improved nutrient absorption and overall digestive efficiency in broilers.

Between the BR1 and BR6 groups, ritalinic acid, γ-aminobutyric acid, N’-formylkynurenine, 7-ketodeoxycholic acid, oxaceprol, 4-hydroxybenzaldehyde, L-dihydroorotic acid, N-acetylcadaverine, PC (8:0/8:0) and thymidine were down-regulated in the BR6 groups, while FFA (20:1), cis-EODA, 1-methylguanosine, 2′-O-methylguanosine, heparin, 1-methylinosine, quinoline-4-carboxylic acid, 2-methylguanosine, arachidyl glycine and 2-(dimethylamino) guanosine were up-regulated in the BR6 groups (Figure 1D). The down-regulation of metabolites such as GABA, bile acids and nucleosides in broilers fed stored brown rice suggests potential disruptions in gut health, including reduced anti-inflammatory capacity, compromised gut barrier function and altered lipid metabolism [54,55,56]. Additionally, decreased GABA levels might relieve suppression of the NF-κB pathway, exacerbating intestinal inflammation and oxidative stress [57]. However, this hypothesis requires further validation through detection of inflammatory factors. Conversely, the up-regulation of FFA (20:1)-modified nucleosides and compounds involved in immune modulation indicates possible adaptive responses to maintain gut homeostasis [58,59]. These findings highlight the importance of lipid oxidation of stored brown rice that influence the gut health and overall metabolic function of broilers.

The K-means clustering in Figure 2B further categorizes the different metabolites into subclasses, highlighting the complexity of metabolic alterations in the ileum of broilers due to the long-term storage of brown rice. Figure 2C,D present the differentially abundant metabolites and their associated metabolic pathways. Notably, pathways like linoleic acid metabolism, one carbon pool by folate, and nicotine and nicotinamide metabolism were enriched in the BR1 group compared with the Corn group; these are crucial for lipid metabolism and energy production [60]. In contrast, the BR1 group vs. the BR6 group comparison revealed significant changes in pathways like drug metabolism-cytochrome P450, GnRH secretion, and α-linolenic acid metabolism. The up-regulation of cytochrome P450 pathways in the BR1 group indicates the enhanced detoxification capabilities, while changes in GnRH secretion could affect hormonal balance and growth performance [61]. These findings underscore the potential metabolic impact of lipid oxidation of stored brown rice on broiler ileum, which could influence digestive efficiency, nutrient absorption, and overall health. Further research is needed to explore the functional implications of these metabolic changes and understand how they might impact the health benefits and quality of broiler products derived from diets containing stored grains.

## 4. Materials and Methods

### 4.1. Animal Ethics Statement

All experiments adhered to the Chinese Guidelines for Animal Welfare and Experimental Protocol, and were approved in advance by the Animal Care and Use Committee of the Academy of National Food and Strategic Reserves Administration (Approval No. 20241221002; date: 21 December 2024).

### 4.2. Experimental Materials

Paddy rice stored for periods of 1 or 6 years was obtained from the National Grain Reserve Barn in Heilongjiang Province. A constant storage temperature of 20 °C was maintained in the grain barn throughout the year. The paddy rice was firstly processed to obtain brown rice, which was then used for the subsequent experimental procedures.

### 4.3. Lipid Metabolism Indexes and Fatty Acid Composition of BR6

The fatty acid value was measured using the procedures specified in GB/T 20569–2006 (Guidelines for evaluation of paddy storage character) [62]. MDA is commonly assessed via the thiobarbituric acid (TBA) reaction method, which entails the creation of a pink complex that is detectable by spectrophotometry. Detecting carbonylated proteins depends on reacting carbonyl groups with 2,4-dinitrophenylhydrazine (DNPH), and subsequent immunoblotting with anti-DNP antibodies. CAT, SOD, T-AOC, GSH and GSH-PX were determined by assay kits of Nanjing Jiancheng Bioengineering Institute (Nanjing, China).

The fatty acid composition of brown rice was determined using the internal standard method according to GB 5009. 168–2016 [63]. Briefly, 0.2 g rice flour was placed into one 50 mL centrifuge tube containing 6 mL of extraction solvent (2 mL 95% ethanol and 4 mL water), 100 mg of pyrogallic acid to prevent oxidation, and 2 mL of tridecanoin (internal standard solution). Then, 10 mL of hydrochloric acid solution was added and the mixture was hydrolyzed at 70–80 °C for 40 min with intermittent shaking every 10 min. Once cooled to room temperature, 2 mL of hexane was added, and the mixture was centrifuged at 4 °C and 1000 rpm for 5 min. Finally, 600 µL of the supernatant was injected into the gas chromatograph, which was equipped with a high-polarity capillary column (e.g., Agilent HP-5, 30 m × 0.32 mm inner diameter, 0.25 µm film thickness). The initial column temperature was set to 150 °C, and the temperature and flow rate were adjusted to separate and detect different fatty acids.

### 4.4. Animals and Dietary Treatments

A total of 240 one-day-old Ross 308 male chicks were weighed at day 1 and then allocated to their respective pens based on a completely randomized design, ensuring that each pen had a similar initial body weight (±20 g/pen) across treatments. The location of the animals and cages were also randomly assigned using Excel’s randomization function. Based on the feasibility of the experimental design and the experience of the preliminary study, each of the treatments had 8 replicate pens of 10 birds during the study. Broilers were housed in floor pens with dimensions of 0.725 m × 1.175 m (0.85 m^2^), using softwood shavings as litter at a depth of 8 cm. The housing environment was controlled, and feed and water were provided *ad libitum* throughout the study.

The experiment utilized a completely randomized design, comprising one control diet (Corn) and two experimental diets. The diets were formulated by completely replacing corn with brown rice stored for either 1 year (BR1) or 6 years (BR6), and in accordance with the Feeding Standard for Chickens (NY/T 33–2004) [64]. The experiment consisted of two stages: I stage (1–21 days of age) and II stage (22–42 days of age) (Table 6).

### 4.5. Performance Measurement and Sampling

The initial body weight of the chicks was recorded on the first day of the experiment. At 21 and 42 days of age, the body weights of all broilers were measured at 07: 00 on the following day after 12 h of fasting. Throughout the experiment, feed intake was monitored and recorded for each replicate. Based on these measurements, the ADG, ADFI, and FCR were calculated for each stage of the experiment. After weighing at 42 days of age, one broiler with a weight close to the average was selected from each cage. A sharp and sterilized blade was used to swiftly sever the head from the body at the neck to ensure immediate death, performed by a trained technician. Left pectoral muscle and ileum tissues were collected for meat quality and enzyme activity determination (*n* = 8) and metabolomics analysis (*n* = 6). In order to improve the reliability of the results, we used a partially blind design in animal experiments, where only the experimental operator was aware of the grouping, and the results evaluators and data analysts remained in the dark until data analysis was completed.

The meat quality of broilers was assessed according to the guidelines of NY/T 1333–2007 [65]. The pH levels of the muscle were measured at three different points for 24 h. The left pectoral muscles were then cut into blocks measuring 5 cm × 3 cm × 2 cm and weighed (marked as m1). These samples were hung in a closed plastic bag at 4 °C for 24 h and then weighed again after using filter paper to absorb surface moisture (marked as m2). The cooking percentage was determined by taking approximately 30 g samples from the left breast muscle (recorded as m3), boiling in boiling water for 30 min, hanging in a cool place for 30 min, and then weighing after absorbing surface moisture with filter paper (recorded as m4). Additionally, about 20 g of meat sample was lyophilized to determine the fatty acid profile using the internal standard method according to GB 5009. 168–2016, as mentioned before [64]. Drip loss and cooking percentage were calculated using the following formula:Drip loss (%) = [(m1 − m2)/m1] × 100;Cooking percentage (%) = (m4/m3) × 100

### 4.6. Antioxidant Properties Analyses

Ileum samples underwent homogenization in a cold maleic acid buffer (0.1 mol/L, pH 6.8) at a ratio of 1:10 (*w*/*v*), then they were subsequently centrifuged at 3000× *g* for 10 min. The supernatants were collected for analysis. The level of MDA and the activities of SOD, T-AOC, GSH-Px and CAT were evaluated using ELISA kits by Nanjing Jiancheng Bioengineering Institute (Nanjing, China). Additionally, the activities of several digestive enzymes, including α-amylase, trypsin, chymotrypsin, and lipase, were determined following the protocols provided in assay kits by Nanjing Jiancheng Bioengineering Institute (Nanjing, China) [66].

### 4.7. Untargeted Metabolome Analysis of Broiler Ileum

#### 4.7.1. Extraction of Metabolites

In total, 50 mg of the sample was precisely weighed and transferred into a 2 mL centrifuge tube. Subsequently, 2 grinding beads and 400 µL of an extraction solution (comprising methanol and water in a 4:1 volume ratio) were added to the tube, with an 0.02 mg/mL of internal standard (L-2-chlorophenylalanine). The samples were then subjected to grinding in a frozen tissue grinder for 6 min at −10 °C and a frequency of 50 Hz. Following the grinding process, the samples were treated with low-temperature ultrasonic extraction for 30 min at 5 °C and 40 kHz. After this step, the samples were kept at −20 °C for 30 min and then centrifuged at 13,000× *g* and 4 °C for 15 min. The supernatant was carefully collected and transferred to an injection vial for subsequent LC-MS/MS analysis [67].

To ensure the stability and reliability of the analytical process, a quality control (QC) sample was prepared by pooling equal volumes from all samples. This QC sample was injected at regular intervals (every 5 to 15 samples) throughout the experiment to monitor the consistency of the analysis.

#### 4.7.2. Liquid Chromatography and Mass Spectrum Parameters

The sample extracts were analyzed using a UPLC-MS/MS system, which included an ExionLC AD UPLC and a TripleTOF 6600 MS (AB SCIEX, Marlborough, MA, USA). Chromatographic separation was performed on an HSS T3 C18 column (100 mm × 2.1 mm i.e., 1.8 µm particle size; Waters, Milford, MA, USA). The mobile phase was composed of a binary solvent system: water (with 0.1% formic acid) and acetonitrile (also with 0.1% formic acid). The gradient elution program was set as follows: 0–10, 95:5 (*v*/*v*) at 0 min, 10:90 (*v*/*v*) at 10.0 min, 10:90 (*v*/*v*) at 11.0 min, 95:5 (*v*/*v*) at 11.1 min, 95:5 (*v*/*v*) at 14 min, then holding this condition until the end of the run at 14.0 min. The column temperature was kept at 40 °C, the flow rate was set at 0.4 mL/min, and the injection volume was 2 µL.

The TripleTOF mass spectrometer was utilized for its ability to conduct MS/MS scans in information-dependent acquisition (IDA) mode during LC/MS analyses. In this mode, the acquisition software (TripleTOF 6600, AB SCIEX, Marlborough, MA, USA) evaluates the full-scan MS data in real-time and initiates MS/MS spectra acquisition based on predefined criteria. During each acquisition cycle, up to 12 precursor ions with an intensity greater than 100 were chosen for fragmentation at a collision energy of 30 V. Each MS/MS event had a product ion accumulation time of 50 milliseconds. The ESI source conditions were set as follows: ion source gas 1 at 50 Psi, ion source gas 2 at 50 Psi, curtain gas at 25 Psi, source temperature at 500 °C, and ion spray voltage floating (ISVF) at 5500 V in positive mode or −4500 V in negative mode.

#### 4.7.3. Data Extraction and Processing

Raw data preprocessing was performed using Progenesis QI (Waters Corporation, Milford, MA, USA), which generated a three-dimensional CSV data matrix that included sample information, metabolite names, and their corresponding mass spectral response intensities. Metabolite identification was supported by a combination of the self-built target compound database MWDB (containing secondary spectra and retention time, RT) and the integrated public database MHK (encompassing Metlin, HMDB, KEGG databases, secondary spectra, and RT), as well as MetDNA, with analysis conducted on the Metware cloud platform.

Metabolic features detected in at least 80% of the samples were retained. After filtering, metabolite values below the quantitation limit were imputed, and the features were normalized by summation to correct for errors arising from sample preparation and instrument variability. The mass spectral peak intensities were normalized using the sum normalization method, resulting in a refined data matrix. Variables with a relative standard deviation (RSD) greater than 30% in QC samples were removed, and log10 transformation was applied to prepare the final data matrix for subsequent analysis. Variance analysis was conducted on the preprocessed data matrix, followed by the application of the R package “ropls” (Version 1.6.2) for principal component analysis (PCA) and OPLS-DA. Metabolites with a VIP score > 1 and *p*-value < 0.05, as determined by the OPLS-DA model and Student’s *t*-test, were identified as significantly different. Differential metabolites were then mapped to their respective biochemical pathways using metabolic enrichment and pathway analysis based on the KEGG database. Enrichment analysis, which evolved from single metabolite annotation to group annotation, was conducted using the Python package “scipy.stats” (Version 1.11.3) to discern the most pertinent biological pathways influenced by the experimental treatments.

Variance analysis was performed on the preprocessed data matrix. Subsequently, the R package “ropls” (Version 1.6.2) was utilized to conduct principal component analysis (PCA) and orthogonal partial least squares discriminant analysis (OPLS-DA). Metabolites identified as significantly different were those with a VIP score exceeding 1 and a *p*-value less than 0.05, as determined by the OPLS-DA model and Student’s t-test. These differential metabolites were then mapped to their corresponding biochemical pathways through metabolic enrichment and pathway analysis, using the KEGG database as a reference. To identify the most relevant biological pathways affected by the experimental treatments, enrichment analysis was carried out using the Python package “scipy.stats” (Version 1.11.3), progressing from single metabolite annotation to group annotation.

### 4.8. Data Analysis

Data are presented as the mean ± SEM. The PROC UNIVERSATE program (SAS Inst. Inc., Carry, NC, USA) in SAS 9.2 was employed to assess the normality and identify any outliers in the values of lipid metabolism indexes and fatty acid composition of BR6, as well as growth performance, meat quality, fatty acid composition, antioxidant, and digestive enzyme indices of broilers. For statistical analysis, lipid metabolism indexes and fatty acid composition of brown rice were evaluated using independent sample tests. In contrast, growth performance, meat quality, fatty acid composition, antioxidant, and digestive enzyme indices of broilers were analyzed using one-way ANOVA, followed by the Student–Newman–Keuls post hoc test in SAS. A *p*-value of less than 0.05 was considered to indicate statistical significance.

## 5. Conclusions

The present study indicates that replacing corn with brown rice has no effect on growth performance or meat quality of broilers. However, long-term storage caused changes in fatty acid composition of brown rice and accelerated lipid oxidation. These changes further altered fatty acid composition of broiler breast muscle and reduced the activity of digestive enzymes in the ileum. Furthermore, metabolomics analysis revealed that long-term storage brown rice may affect broiler intestinal metabolic characteristics through pathways involving GABA, bile acid and nucleotide metabolism.

## Figures and Tables

**Figure 1 ijms-26-07025-f001:**
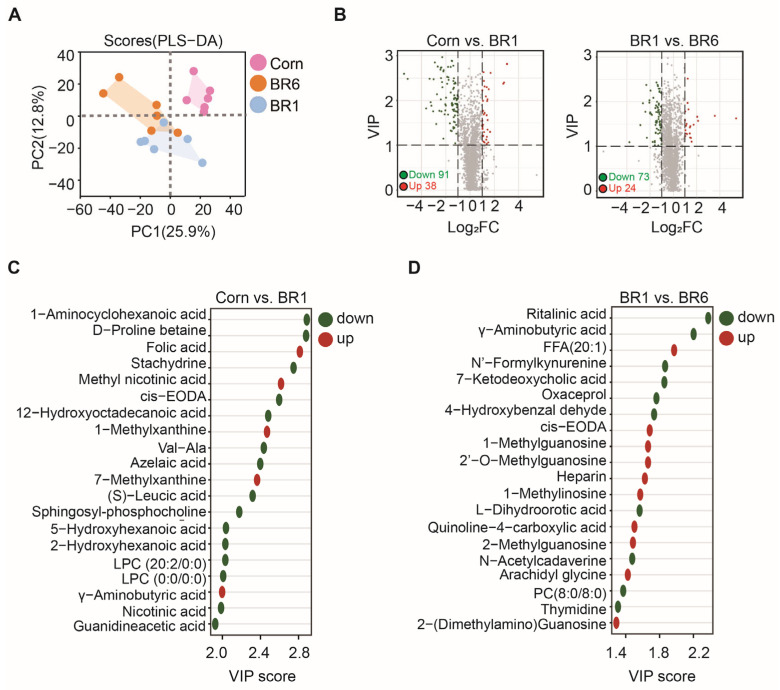
PLS-DA score plots and significantly changed metabolites in the ileum of broilers fed corn- and brown rice-based diets. (**A**) PLS-DA score plot; (**B**) volcano map; (**C**) significantly changed metabolites between the Corn and BR1 groups; (**D**) significantly changed metabolites between the BR1 and BR6 groups. Note: Corn, corn-based diet group; BR6, stored brown rice-based diet group; BR1, fresh brown rice-based diet group; PLS-DA, partial least squares discriminant analysis; VIP, variable importance in the projection. *n* = 6.

**Figure 2 ijms-26-07025-f002:**
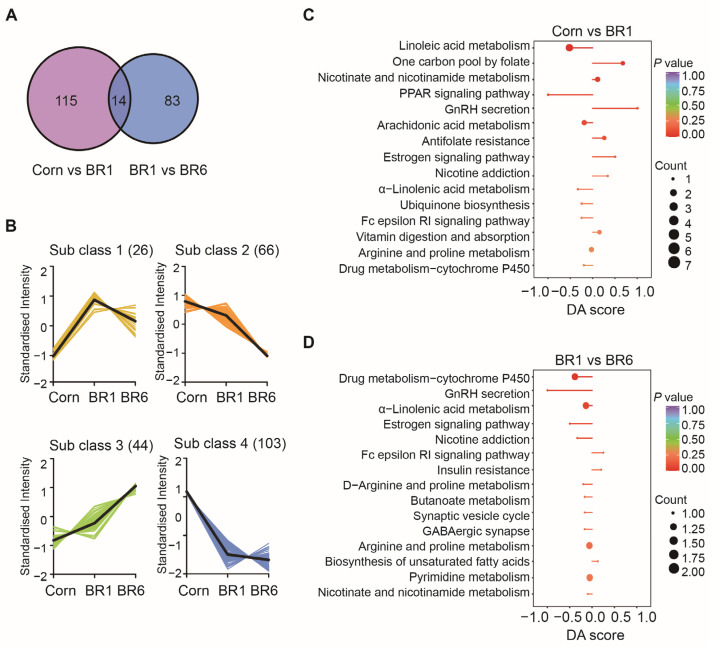
Significantly changed metabolic pathways in ileum of broilers fed corn- and brown-rice-based diets. (**A**) Venn diagram; (**B**) K-means clustering analysis; (**C**) significantly changed metabolic pathways between the Corn and BR1 groups; (**D**) significantly changed metabolic pathways between the BR1 and BR6 groups. Note: Corn, corn-based diet group; BR6, stored brown rice-based diet group; BR1, fresh brown rice-based diet group; DA score, The differential abundance score. *n* = 6.

**Table 1 ijms-26-07025-t001:** Lipid metabolism indexes of brown rice.

Items	BR6	BR1	*p*-Value
Fatty acid value (mg KOH/100 g)	27.83 ± 2.11 ^a^	17.44 ± 1.05 ^b^	<0.01
MDA (nmol/mg)	6.24 ± 0.54 ^a^	3.67 ± 0.30 ^b^	<0.01
Carbonylated protein (nmol/mg)	3.25 ± 0.32 ^a^	1.60 ± 0.05 ^b^	<0.01
SOD (U/mg)	6.48 ± 0.93	6.52 ± 0.52	0.97
GSH-PX (U/mg)	41.80 ± 5.45 ^b^	68.13 ± 5.87 ^a^	<0.01
GSH (μmol/mg)	1.52 ± 0.06 ^b^	2.18 ± 0.18 ^a^	<0.01
CAT (U/mg)	7.44 ± 0.86	8.40 ± 0.67	0.40
LPS (U/mg)	2.66 ± 0.33	1.86 ± 0.36	0.13
APX (U/mg)	0.94 ± 0.10	1.25 ± 0.15	0.11

Note: BR6, stored brown rice; BR1, fresh brown rice; MDA, malondialdehyde; CAT, catalase; SOD, superoxide dismutase; GSH-Px, glutathione peroxidase; GSH, glutathione; CAT, catalase; LPS, lipase; APX, ascorbate peroxidase. Results are expressed as mean ± standard error of the mean (SEM), *n* = 6. ^a,b^ Peer data with different shoulder labels (lowercase English letters) indicate significant differences (*p* < 0.05).

**Table 2 ijms-26-07025-t002:** Fatty acid composition of brown rice (%).

Items	BR6	BR1	*p*-Value
Myristic acid (C14: 0)	0.55 ± 0.02	0.54 ± 0.01	0.75
Palmitic acid (C16: 0)	25.40 ± 0.07	25.58 ± 0.17	0.27
Stearic acid (C18: 0)	2.13 ± 0.06 ^a^	1.86 ± 0.02 ^b^	<0.01
Oleic acid (C18: 1n-9c)	33.98 ± 0.18 ^a^	31.74 ± 0.44 ^b^	<0.01
Linoleic acid (C18: 2n-6c)	33.88 ± 0.27 ^b^	36.01 ± 0.25 ^a^	<0.01
Alpha-linolenic acid (C18: 3n-3)	1.07 ± 0.02 ^b^	1.46 ± 0.03 ^a^	<0.01
Icosanoic acid (C20: 0)	0.64 ± 0.02 ^a^	0.56 ± 0.00 ^b^	<0.01
Eicosenoic acid (C20: 1)	0.55 ± 0.01 ^a^	0.50 ± 0.01 ^b^	<0.01
Docosanoic acid (C22: 0)	0.33 ± 0.01 ^a^	0.30 ± 0.00 ^b^	0.03
Docosahexaenoic acid (C22: 6n-3)	0.33 ± 0.02	0.37 ± 0.01	0.79
Tricosanoic acid (C23: 0)	0.30 ± 0.01 ^b^	0.36 ± 0.01 ^a^	0.04
Tetracosanoic acid (C24: 0)	0.84 ± 0.04 ^a^	0.72 ± 0.01 ^b^	0.03
Total (g/100g)	2.91± 0.16	2.69± 0.03	0.23
SFA	30.16 ± 0.18	29.91 ± 0.19	0.38
MUFA	34.53 ± 0.19 ^a^	32.24 ± 0.45 ^b^	<0.01
PUFA	35.32 ± 0.28 ^b^	37.85 ± 0.29 ^a^	<0.01
UFA	69.84 ± 0.18	70.09 ± 0.19	0.38
SFA/UFA	0.43 ± 0.00	0.43 ± 0.00	0.38
PUFA/SFA	1.17 ± 0.02 ^b^	1.27 ± 0.01 ^a^	<0.01

Note: BR6, stored brown rice; BR1, fresh brown rice; SFA, saturated fatty acid; MUFA, monounsaturated fatty acid; PUFA, polyunsaturated fatty acid; UFA, unsaturated fatty acid; EFA, essential fatty acid. Results are expressed as mean ± SEM, *n* = 6. ^a,b^ Peer data with different shoulder labels (lowercase English letters) indicate significant differences (*p* < 0.05).

**Table 3 ijms-26-07025-t003:** Growth performance and meat quality of broilers.

Items	Corn	BR6	BR1	*p*-Value
Growth performance				
Feed intake, kg				
Wk 0 to 3	36.24 ± 0.42	36.32 ± 0.85	35.22 ± 0.66	0.44
Wk 4 to 6	98.45 ± 3.33	93.70 ± 4.64	94.77 ± 3.97	0.69
Wk 0 to 6	67.36 ± 1.78	64.95 ± 2.08	64.95 ± 2.26	0.64
Body weight gain, kg				
Wk 0 to 3	27.60 ± 0.29	26.65 ± 0.57	26.72 ± 0.47	0.29
Wk 4 to 6	63.65 ± 1.31	62.51 ± 0.92	64.85 ± 0.65	0.28
Wk 0 to 6	45.64 ± 0.74	44.58 ± 0.42	45.79 ± 0.54	0.30
FCR				
Wk 0 to 3	1.31 ± 0.01 ^b^	1.36 ± 0.01 ^a^	1.32 ± 0.01 ^b^	0.03
Wk 4 to 6	1.55 ± 0.05	1.50 ± 0.06	1.46 ± 0.06	0.56
Wk 0 to 6	1.48 ± 0.03	1.47 ± 0.04	1.42 ± 0.04	0.58
Meat quality				
pH	5.63 ± 0.02	5.58 ± 0.02	5.62 ± 0.01	0.13
Drop loss, %	5.56 ± 0.28 ^b^	7.00 ± 0.25 ^a^	5.93 ± 0.36 ^b^	<0.01
Cooking percentage, %	63.48 ± 0.96	60.53 ± 0.92	61.58 ± 0.55	0.06

Note: Corn, corn-based diet group; BR6, stored brown rice-based diet group; BR1, fresh brown rice-based diet group; FCR, feed conversion ratio. Results are expressed as mean ± SEM, *n* = 8. ^a,b^ Peer data with different shoulder labels (lowercase English letters) indicate significant differences (*p* < 0.05).

**Table 4 ijms-26-07025-t004:** Fatty acid composition in broiler breast muscle (%).

Items	Corn	BR6	BR1	*p*-Value
Myristic acid (C14: 0)	0.55 ± 0.02 ^a,b^	0.61 ± 0.06 ^a^	0.49 ± 0.02 ^b^	0.09
Palmitic acid (C16: 0)	22.91 ± 0.39 ^b^	24.48 ± 0.23 ^a^	24.39 ± 0.18 ^a^	<0.01
Palmitoleic acid (C16: 1)	1.29 ± 0.09 ^b^	2.54 ± 0.14 ^a^	2.79 ± 0.08 ^a^	<0.01
Stearic acid (C18: 0)	14.37 ± 0.37 ^a^	12.75 ± 0.36 ^b^	11.90 ± 0.19 ^b^	<0.01
Oleic acid (C18: 1n-9c)	20.59 ± 0.54 ^b^	26.98 ± 1.13 ^a^	28.49 ± 0.31 ^a^	<0.01
Linoleic acid (C18: 2n-6c)	23.87 ± 0.60 ^a^	19.56 ± 0.37 ^b^	20.03 ± 0.34 ^b^	<0.01
Alpha-linolenic acid (C18: 3n-3)	1.05 ± 0.07	0.96 ± 0.06	1.15 ± 0.04	0.09
Eicosenoic acid (C20: 1)	0.54 ± 0.03	0.47 ± 0.04	0.34 ± 0.05	0.13
Eicosandienoic acid (C20: 2)	1.32 ± 0.10 ^a^	0.79 ± 0.06 ^b^	0.71 ± 0.04 ^b^	<0.01
Sciadonic acid (C20: 3n-6)	1.28 ± 0.09	1.34 ± 0.13	1.0 ± 0.06	0.17
Docosadienoic acid (C20: 4n-6)	8.52 ± 0.51 ^a^	6.02 ± 0.48 ^b^	5.36 ± 0.16 ^b^	<0.01
Erucic Acid (C22: 1n-9)	2.51 ± 0.28	2.48 ± 0.32	0.33 ± 0.21	0.89
Nervonic acid (C24: 1)	0.55 ± 0.02 ^a^	0.54 ± 0.05 ^a^	0.43 ± 0.02 ^b^	0.04
Docosahexaenoic acid (C22: 6n-3)	0.67 ± 0.05 ^a^	0.47 ± 0.04 ^b^	0.48 ± 0.02 ^b^	<0.01
Total (mg/g)	0.61± 0.02 ^b^	0.84 ± 0.07 ^a^	0.92 ± 0.05 ^a^	<0.01
SFA	37.82 ± 0.63	37.5 ± 0.36	36.78 ± 0.27	0.18
MUFA	25.48 ± 0.62 ^b^	33.01 ± 0.92 ^a^	34.33 ± 0.33 ^a^	<0.01
PUFA	36.70 ± 0.27 ^a^	29.1 ± 0.68 ^b^	28.83 ± 0.50 ^b^	<0.01
UFA	62.18 ± 0.63	62.15 ± 0.36	63.22 ± 0.7	0.18
SFA/UFA	0.61 ± 0.02	0.61 ± 0.01	0.58 ± 0.01	0.18
PUFA/SFA	0.97 ± 0.02 ^a^	0.77 ± 0.02 ^b^	0.78 ± 0.02 ^b^	<0.01
MUFA/PUFA	0.69 ± 0.02 ^b^	1.14 ± 0.06 ^a^	1.20 ± 0.03 ^a^	<0.01
EFA	24.92 ± 0.66 ^a^	20.52 ± 0.38 ^b^	21.19 ± 0.37 ^b^	<0.01

Note: Corn, corn-based diet group; BR6, stored brown rice-based diet group; BR1, fresh brown rice-based diet group; SFA, saturated fatty acid; MUFA, monounsaturated fatty acid; PUFA, polyunsaturated fatty acid; UFA, unsaturated fatty acid; EFA, essential fatty acid. Results are expressed as mean ± SEM, *n* = 8. ^a,b^ Peer data with different shoulder labels (lowercase English letters) indicate significant differences (*p* < 0.05).

**Table 5 ijms-26-07025-t005:** Ileal antioxidant and digestive enzyme activities of broilers.

Items	Corn	BR6	BR1	*p*-Value
MDA	0.91 ± 0.02 ^a^	0.62 ± 0.01 ^b^	0.51 ± 0.01 ^c^	<0.01
SOD (U/mg)	10.52 ± 0.35 ^b^	11.73 ± 0.42 ^a^	12.42 ± 0.27 ^a^	<0.01
TAOC (U/mg)	1.99 ± 0.02 ^b^	2.72 ± 0.08 ^a^	2.89 ± 0.10 ^a^	<0.01
GSH-PX (U/mg)	266.12 ± 8.32 ^b^	305.07 ± 13.29 ^a^	308.06 ± 9.73 ^a^	0.02
CAT (U/mg)	9.64 ± 0.47 ^b^	13.04 ± 0.29 ^a^	13.81 ± 0.29 ^a^	<0.01
α-AMY (U/mg)	1.11 ± 0.03 ^a^	0.84 ± 0.02 ^b^	1.12 ± 0.04 ^a^	<0.01
Trypsin (U/mg)	39.01 ± 1.80 ^a^	27.40 ± 1.54 ^b^	42.96 ± 1.21 ^a^	<0.01
Chymotrypsin (U/mg)	70.00 ± 2.71 ^a^	55.37 ± 0.94 ^b^	74.65 ± 4.16 ^a^	<0.01
LPS (U/mg)	50.94 ± 0.94 ^b^	42.69 ± 1.57 ^c^	56.30 ± 2.34 ^a^	<0.01

Note: Corn, corn-based diet group; BR6, stored brown rice-based diet group; BR1, fresh brown rice-based diet group; MDA, malonaldehyde; SOD, superoxide dismutase; T-AOC, total antioxidant capacity; GSH-Px, glutathione peroxidase; CAT, catalase; AMY, amylase; LPS, lipase. Results are expressed as mean ± SEM, *n* = 8. ^a,b^ Peer data with different shoulder labels (lowercase English letters) indicate significant differences (*p* < 0.05).

**Table 6 ijms-26-07025-t006:** Composition and nutrient levels of experimental diets (%, as-fed basis).

Items	1–21 Days of Age			22–42 Days of Age		
	Corn	BR6	BR1	Corn	BR6	BR1
Ingredients						
Corn	55.88	/	/	60.74	/	/
Soybean meal	34.76	35.04	35.90	27.48	27.90	27.90
BR6	0.00	56.97	/	0.00	62.51	/
BR1	0.00	/	56.68	0.00	/	62.76
Corn gluten meal	3.00	2.22	1.50	4.42	4.12	3.83
Soybean oil	3.28	1.90	2.10	3.68	1.98	2.00
CaHPO4	1.78	1.68	1.68	1.48	1.38	1.38
Limestone	1.26	1.32	1.32	1.24	1.30	1.32
NaCl	0.30	0.30	0.30	0.30	0.30	0.30
Vitamin premix ^a^	0.03	0.03	0.03	0.03	0.03	0.03
Mineral premix ^b^	0.20	0.20	0.20	0.20	0.20	0.20
Choline chloride (50%)	0.10	0.10	0.10	0.10	0.10	0.10
Met	0.21	0.17	0.17	0.14	0.10	0.10
Lys-HCl (98%)	0.16	0.05	0.02	0.17	0.07	0.07
Thr	0.04	0.02	0.00	0.02	0.01	0.01
Total	100	100	100	100	100	100
Nutrients levels ^c^						
Dry matter	89.05	88.91	89.67	88.70	89.10	88.63
Crude protein	21.59	21.40	21.46	20.16	20.07	19.92
Calcium	1.03	1.02	1.05	0.90	0.92	0.89
Total phosphorus	0.69	0.70	0.68	0.67	0.66	0.66
Metabolizable energy, MJ/kg	12.48	12.48	12.48	12.89	12.89	12.89

^a^ Vitamin premix provided the following per kg of diet: vitamin A, 8000 IU; vitamin D3, 1000 IU; vitamin E, 20 IU; vitamin K3, 0.5 mg; vitamin B12, 10 µg; vitamin B2, 9.6 mg; vitamin B1, 2 mg; vitamin B5, 20 mg; vitamin B6, 3.5 mg; niacin acid, 35 mg; calcium pantothenate, 10 mg; folic acid, 0.55 mg; biotin, 0.18 mg; ^b^ Mineral premix provided the following per kg of diet: Mn, 100 mg (as manganese oxide); Fe, 80 mg (as ferrous sulfate); Zn, 80 mg (as zinc oxide); Cu, 8 mg (as copper sulfate); I, 0.7 mg (as ethylenediamine dihydroiodide) and Se, 0.3 mg (as sodium selenite).^c^ Metabolizable energy were calculated, other indicators were measured.

## Data Availability

Raw metabolomics data were uploaded to the MetaboLights database (accession number: MTBLS12604; https://www.ebi.ac.uk/metabolights/MTBLS12604, accessed on 14 June 2025; European Bioinformatics Institute, Cambridge, UK).

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
