# Peer review of "Lipid Oxidation of Stored Brown Rice Changes Ileum Digestive and Metabolic Characteristics of Broiler Chickensâ€"

_ijms, 2025, doi:10.3390/ijms26147025_

Round 1
Reviewer 1 Report
Comments and Suggestions for Authors
Dear authors,
Please see attached.

Author Response
Comments 1: Although the abstract mentions "few pay attention to the changes in animal physiological metabolism caused by lipid oxidation of grains," elaborating on this in the introduction would provide a clearer rationale for the metabolomics approach. Additionally, a brief mention of the anticipated practical implications of this research for the poultry industry in the introduction would immediately highlight its significance to a broader audience.
Response 1: Thank you very much for your careful review and for bringing the issue to our attention. (1) We have elaborated in detail on the research on the impact of grain oxidation on the physiological and metabolic changes of animals, particularly emphasizing the significant role of metabolomics in such studies (Lines 65-78) (2) We added the anticipated practical implications of this research in Lines 85-90 of the manuscript.
Comments 2: In the Results section, particularly concerning the growth performance and meat quality data, while the authors state that "most parameters did not show significant differences," a more nuanced discussion of the trends observed, even if not statistically significant, could be beneficial. For instance, the slightly lower FCR in the corn diet group during the first three weeks is noted, but further speculation on why this trend might exist, or a statement about the practical significance of such small differences in a commercial setting, would add depth. For the meat quality, the significant difference in drip loss and the trend in cooking percentage are important findings, and perhaps a more direct link to the implications for consumer acceptance or processing quality could be drawn in the results or discussion.
Response 2: Thank you very much for your review. (1) We have addressed your concern regarding the non-significant differences by providing enhanced discussion in Lines 261-267 of the revised manuscript:”Although FCR difference was marginal in this trial period, large-scale application could yield substantial cost and feed savings. The freshness of feed has a significant impact on the early growth performance of broilers.” (2)Regarding drip loss and cooking yield, we have expanded our discussion in Lines 275-280 to address meat quality attributes including color, texture, and consumer acceptance: “Meat with higher drip loss tends to lose moisture during storage and retail display, resulting in dry surface and dull coloration. The poor water-holding capacity also adversely affects tenderness and juiciness[60]. Meanwhile, meat with high cooking loss exhibits inferior chewing properties and excessive firmness during processing, directly impairing eating quality and consequently reducing consumer purchasing intention.”
Comments 3: When discussing the fatty acid composition of broiler breast muscle, the authors acknowledge that corn's fatty acid composition was not determined. While this is acceptable, a brief statement on how this limitation might impact the interpretation of the observed differences in broiler muscle fatty acid profiles, or a suggestion for future research to include such data, would demonstrate greater transparency. Similarly, while the discussion links the observed metabolic changes to potential impacts on gut health, a more explicit hypothesis on the specific mechanisms through which altered metabolites like GABA or bile acids exert their effects in the broiler ileum could enrich the scientific narrative.
Response 3: Thank you very much for your review. Regarding your concern about "the limitation of not determining corn's fatty acid composition for interpreting broiler muscle fatty acid profiles," we have added the following clarification in Lines 307-310: “Due to the lack of data on the fatty acid composition of corn, it is difficult to fully de-termine the contribution of corn to the changes in the fatty acid composition of broiler muscle. Future research should determine the fatty acid composition of corn to gain a more comprehensive understanding of its impact on the fatty acid composition of broilers.”(2)We added hypotheses on the mechanism by which changes in GABA levels affect the ileal function of broilers in lines 402-407:”The down-regulation of GABA levels in the BR6 group may release its inhibitory effect on intestinal motility, accelerating peristalsis and consequently reducing nutrient absorption time and efficiency[62]. Additionally, decreased GABA levels might relieve suppression of the NF-κB pathway, exacerbating intestinal inflammation and oxida-tive stress[63], However, this hypothesis requires further validation through detection of inflammatory factors.”
Comments 4: In terms of overall presentation, ensuring consistent terminology throughout the manuscript, particularly for the brown rice treatments (e.g., consistently using BR1 and BR6 rather than sometimes referring to them as "fresh brown rice" and "stored brown rice" interchangeably without explicit context), would improve readability. Furthermore, while the figures are informative, ensuring that all abbreviations within the figures are explicitly defined in the figure legends or the main text upon their first appearance would prevent readers from having to search for their meaning.
Response 4: Thank you for your valuable feedback. We have revised the manuscript to ensure consistent terminology for the brown rice treatments, using BR1 and BR6 consistently throughout. We also explicitly defined all abbreviations in the figure legends or main text upon their first appearance to enhance readability.

Reviewer 2 Report
Comments and Suggestions for Authors
Comments and Suggestions for Authors
After reviewing the manuscript entitled “The lipid oxidation of stored brown rice changed ileum digestive and metabolic characteristics in broiler chickens”, the following suggestions were made it. The manuscript provides interesting information on the effect of oxidized lipids from brown rice on metabolic and digestive parameters in broiler chickens. However, several important changes are required before the manuscript can be considered for publication. The individual corrections are shown below:
Abstract
Lines 13-14: Authors should provide more detailed information about the experimental design. For example, the experimental design used, the average weight (and standard deviation) of the broiler chicks initially used, the genetic strain of the broiler chicks, the number of replicates within each treatment, and the number of broiler chicks within each replicate.
Lines 15-24: The authors should modify (improve) the description of the main results obtained in the study. To improve the description, the authors should add the significance values ​​observed for the different response variables (or groups of response variables). This modification will enable readers to understand better the magnitude of the effect obtained with the treatments.
Keywords: “stored brown rice; lipid oxidation; ileum metabolic characteristics; broiler chickens”. Four of the five keywords used by the authors were previously used in the manuscript title, which is incorrect and should be corrected. The keywords used should be different from the title words to increase the manuscript's identification range in currently available electronic search engines.
Introduction
Lines 33-39: In these lines, the authors present several claims that must be supported by relevant scientific references to verify their scientific validity. Please correct them.
Lines 44-45: Are there any scientific references to support this statement?
Lines 61-63: Authors should complement this information by providing background information (citing some experimental studies) that have evaluated the impact of oxidized fat feed on productive performance, antioxidant status, and intestinal health in broiler chickens and other non-ruminant animals. To ensure the inclusion of relevant background information, each study should specify the level of oxidized fat in the diet, the period of use, and the age or physiological stage of the animals used.
Lines 69-71: These lines should be deleted because they represent a summarized version of the background information presented above. This information should be replaced by a clear hypothesis of the results expected by the authors. This hypothesis should be written after considering the corrections requested in the previous comment.
Line 73: Along these lines, the authors indicate that one of the study's objectives was to evaluate the meat quality of broiler chickens. However, the introduction does not mention any information on the use of oxidized fat in broiler chicken diets and its effect on meat quality. Therefore, the authors should provide background information demonstrating how oxidized fats in the diet negatively impact various physicochemical parameters related to meat quality.
Material and methods
Lines 423-442: All methods used to measure response variables must be justified with a relevant scientific reference. This modification is crucial to verify that the authors have used relevant scientific references.
Lines 482-539: The procedures described for measuring pH and water-holding capacity in meat (drip loss and cooking loss) are replicable. However, the authors should cite the scientific references for the methods used to verify that appropriate methods for broiler meat were used. Similarly, the scientific references for the methods used to measure the antioxidant status of meat, metabolite extraction, liquid chromatography, and mass spectrophotometry should be added by the authors.
Lines 540-585: Appropriate statistical tests were also used to verify the normality of the data. Therefore, I have no additional suggestions in this section.
Results
Table 1: The interpretation and description of this table are correct. However, the meaning of the abbreviations MDA, SOD, GSH-PX, GSH, CAT, LPS, and APX should be added to the table footnote.
Table 2: The interpretation and description of this table are correct. However, the meaning of the abbreviations SFA, MUFA, PUFA, UFA, Corn, BR6 and BR9 should be added to the table footnote. Likewise, the meaning of the superscripts placed above the treatment means must be specified.
Table 4: The interpretation and description of this table are correct. However, the meaning of the abbreviations MDA, SOD, TAOC, GSH-PX, CAT, α-AMY, and LPS should be added to the table footnote.
General Comment: Although the descriptions of the tables and figures shown in the Results section are adequate, a recurring error by the authors is their failure to define the meaning of the abbreviations used within the tables and figures of the results. Therefore, this error should be carefully reviewed throughout the Discussion section and corrected.
Discussion
Lines 212-224: These lines should be removed because they are a description of the results rather than a discussion explaining the observed changes in the response variables.
Lines 235-238: These lines should be removed because they are a description of the results rather than a discussion explaining the observed changes in the response variables.
Lines 242-245: These lines should be removed because they are a description of the results rather than a discussion explaining the observed changes in the response variables. This error is too repetitive; therefore, the authors should review and correct this problem throughout the entire discussion section.
Lines 246-411: In my opinion, the discussion provided by the authors is well-organized and well-written. Furthermore, this section contains enough depth for readers to understand the authors' results adequately. However, the authors should carefully review the entire discussion section to avoid the errors mentioned in the three previous comments. Furthermore, in several discussion paragraphs, the authors mention the table or figure numbers where some results were shown. These citations of tables and figures should be removed from the discussion section because they are incorrect, as they were previously cited in the results section.
Conclusions
Lines 587-590: The conclusions provided by the authors do not represent the totality of the results obtained and need improvement. The authors should conclude on the type of effect (positive or negative) of the treatments on all response variables. In its current form, this section only concludes on the activity of digestive enzymes and metabolic characteristics, which is too ambiguous and unclear. Please correct and clarify.
Author Response
Comments 1: Authors should provide more detailed information about the experimental design. For example, the experimental design used, the average weight (and standard deviation) of the broiler chicks initially used, the genetic strain of the broiler chicks, the number of replicates within each treatment, and the number of broiler chicks within each replicate (Lines 13-14).The authors should modify (improve) the description of the main results obtained in the study. To improve the description, the authors should add the significance values observed for the different response variables (or groups of response variables). (Lines 15-24). Keywords: “stored brown rice; lipid oxidation; ileum metabolic characteristics; broiler chickens”. Four of the five keywords used by the authors were previously used in the manuscript title, which is incorrect and should be corrected. The keywords used should be different from the title words to increase the manuscript's identification range in currently available electronic search engines.
Response 1: Thank you very much for your review. We have supplemented specific experimental design details in the article abstract, enhanced the observed response values for different response variables, and updated the keywords section. We have also made changes to other parts of the Abstract, as detailed in the revised manuscript.
Comments 2:Authors should complement this information by providing background information (citing some experimental studies) that have evaluated the impact of oxidized fat feed on productive performance, antioxidant status, and intestinal health in broiler chickens and other non-ruminant animals. To ensure the inclusion of relevant background information, each study should specify the level of oxidized fat in the diet, the period of use, and the age or physiological stage of the animals used(Lines 61-63).These lines should be deleted because they represent a summarized version of the background information presented above. This information should be replaced by a clear hypothesis of the results expected by the authors. This hypothesis should be written after considering the corrections requested in the previous comment. (Lines 69-71). Along these lines, the authors indicate that one of the study's objectives was to evaluate the meat quality of broiler chickens. However, the introduction does not mention any information on the use of oxidized fat in broiler chicken diets and its effect on meat quality. Therefore, the authors should provide background information demonstrating how oxidized fats in the diet negatively impact various physicochemical parameters related to meat quality (Line 73).
Response 2: Thank you very much for your review.We have supplemented references regarding the effects of oxidized fat on broiler growth performance and antioxidant status in Lines 67-73, while adding discussions on its impact on meat quality with supporting literature in Lines 75-80. Please refer to the revised manuscript for detailed modifications. Thank you again for your valuable comments.
Comments 3: All methods used to measure response variables must be justified with a relevant scientific reference. This modification is crucial to verify that the authors have used relevant scientific references(Lines 423-442).The procedures described for measuring pH and water-holding capacity in meat (drip loss and cooking loss) are replicable. However, the authors should cite the scientific references for the methods used to verify that appropriate methods for broiler meat were used. Similarly, the scientific references for the methods used to measure the antioxidant status of meat, metabolite extraction, liquid chromatography, and mass spectrophotometry should be added by the authors (Lines 482-539).
Response 3: Thank you very much for your review. Following your suggestion, we have annotated and revised the sections lacking references in the Materials and Methods. Please refer to the revised manuscript for detailed modifications. Thank you again for your valuable comments.
Comments 4: In the results, the meaning of the abbreviations should be added to the table footnote.
Response 4: Thank you very much for your review. We also explicitly defined all abbreviations in the figure legends or main text upon their first appearance to enhance readability.
Comments 5: In the discussion, lines 212-224, 235-238, and 242-245 contain repetitive errors where the authors describe results rather than discussing the observed changes, these lines should be removed and the issue be corrected thorough the entire discussion section. Additionally, lines 246-411 contains incorrect citations to tables and figures, which should be removed as they were already cited in the results section.
Response 5: Thank you very much for your review. We have removed the described lines and carefully review the discussion section to ensure it focuses on interpreting the results rather than merely describing them. We also removed any incorrect citations to tables and figures. Pleased see the reviewed manuscript.
Comments 6: The conclusions provided by the authors do not represent the totality of the results obtained and need improvement. The authors should conclude on the type of effect (positive or negative) of the treatments on all response variables. In its current form, this section only concludes on the activity of digestive enzymes and metabolic characteristics, which is too ambiguous and unclear. Please correct and clarify (Lines 587-590).
Response 6: Thank you very much for your review. Based on the findings, we have supplemented the previously missing arguments in conclusions (Lines 609-615). Thank you again for your valuable comments.
